# White matter connectivity in brain networks supporting social and affective processing predicts real-world social network characteristics

Ryan Hyon [1✉], Robert S. Chavez [2], John Andrew H. Chwe[3], Thalia Wheatley[4,5], Adam M. Kleinbaum [6] & Carolyn Parkinson [1,7✉]

Human behavior is embedded in social networks. Certain characteristics of the positions that people occupy within these networks appear to be stable within individuals. Such traits likely stem in part from individual differences in how people tend to think and behave, which may be driven by individual differences in the neuroanatomy supporting socio-affective processing. To investigate this possibility, we reconstructed the full social networks of three graduate student cohorts ($N = 275$; $N = 279$; $N = 285$), a subset of whom ($N = 112$) underwent diffusion magnetic resonance imaging. Although no single tract in isolation appears to be necessary or sufficient to predict social network characteristics, distributed patterns of white matter microstructural integrity in brain networks supporting social and affective processing predict eigenvector centrality (how well-connected someone is to well-connected others) and brokerage (how much one connects otherwise unconnected others). Thus, where individuals sit in their real-world social networks is reflected in their structural brain networks. More broadly, these results suggest that the application of data-driven methods to neuroimaging data can be a promising approach to investigate how brains shape and are shaped by individuals' positions in their real-world social networks.

[1] Department of Psychology, University of California, Los Angeles, Los Angeles, CA, USA. [2] Department of Psychology, University of Oregon, Eugene, OR, USA. [3] Department of Psychology, New York University, New York, NY, USA. [4] Department of Psychological and Brain Sciences, Dartmouth College, Hanover, NH, USA. [5] Santa Fe Institute, Santa Fe, NM, USA. [6] Tuck School of Business, Dartmouth College, Hanover, NH, USA. [7] Brain Research Institute, University of California, Los Angeles, Los Angeles, CA, USA. ✉email: rhyon@ucla.edu; cparkinson@ucla.edu

All human cognition and behavior are embedded within the context of real-world social networks. In any social network, people vary systematically with respect to the number of friends that they have, the extent to which they are well-connected to well-connected others, and the extent to which they connect people who would otherwise be unconnected to each other. Although these social network position characteristics have meaningful consequences for individuals and their communities[1–9], they are not well captured by measures typically used to assess individual differences in personality, such as self-report surveys administered to individuals in isolation[10,11]. However, recent work has shown that social network position characteristics constitute significantly heritable individual difference variables that are stable across contexts[12,13]. The heritability of social network position characteristics may be driven, at least in part, by the heritability of social, affective, and behavioral tendencies, which may be reflected in individual differences in the networks of brain regions that support relevant aspects of social perception, cognition, and affective processing. Here, we sought to investigate this possibility by integrating structural neuroimaging data with characterizations of participants' positions in their real-world social networks.

A growing body of research has begun to highlight the critical role of white matter connectivity in supporting social cognition[14], and the structural integrity of white matter tracts has been linked to a variety of individual differences in social, cognitive, and behavioral traits[15–20]. Although past work has largely focused on how the structural integrity of single white matter tracts relate to sociobehavioral tendencies, using data-driven machine learning models to map relationships between distributed patterns of white matter microstructural integrity and sociobehavioral tendencies can provide an informative window into the complex web of connectivity between brain regions that supports social cognition[21].

Here, we used diffusion magnetic resonance imaging (dMRI) to test whether individual differences in distributed patterns of white matter microstructural integrity are predictive of individual differences in social network position characteristics. To this end, we characterized the complete social networks of three different bounded communities of individuals, a subset of whom underwent dMRI. We then used probabilistic tractography to delineate groups of white matter tracts associated with three key facets of social processing: face perception, mentalizing, and mirroring, as well as affective processing. Finally, we used a machine learning algorithm to predict characteristics of individuals' social network positions based on patterns of white matter microstructural integrity across tracts in these brain networks. Rather than only examining univariate relationships between single tracts and social network position characteristics, leveraging a data-driven, multivariate approach can improve predictive performance by taking into account distributed connectivity signatures (i.e.,multi-tract patterns of white matter microstructural integrity).

Patterns of microstructural integrity distributed across white matter tracts in brain networks involved in social and affective processing were predictive of structural characteristics of individuals' positions in their real-world social networks, such as the extent to which they bridge between disparate people or groups (brokerage) and the extent to which they are well-connected to well-connected others (eigenvector centrality). In addition, while distributed patterns of white matter microstructural integrity were predictive of social network position characteristics, no single white matter tract appeared to be necessary or sufficient for predicting social network position characteristics. These findings suggest that individual differences in brain networks that support social perception, affective processing, and understanding others' actions may be particularly important in determining the structural positions that individuals occupy in their real-world social networks.

## Results and Discussion

We first characterized the complete social networks of three cohorts of a graduate program. Members of each cohort ($N_{cohort-1} = 275$; $N_{cohort-2} = 279$; $N_{cohort-3} = 285$) completed an online survey (see Methods); data from this survey was used to characterize each cohort's social network (Fig. 1). A subset of the individuals who participated in the social network survey also participated in the dMRI study ($N_{cohort-1} = 46$; $N_{cohort-2} = 32$; $N_{cohort-3} = 34$ after exclusions based on participant movement; see Methods), in which diffusion-weighted images were collected (see Methods for more details).

For each dMRI participant, we characterized their position in the social network of their cohort in terms of five social network position characteristics: out-degree centrality (the number of people whom the participant names as a friend), in-degree centrality (the number of people who name the participant as a friend), eigenvector centrality (the extent to which the participant is well-connected to other well-connected individuals), betweenness centrality (a global measure of brokerage measuring the fraction of shortest paths between other members of the social network that pass through the participant), and constraint (a local measure of brokerage accounting for the extent to which someone has access to non-redundant social partners; see Methods for

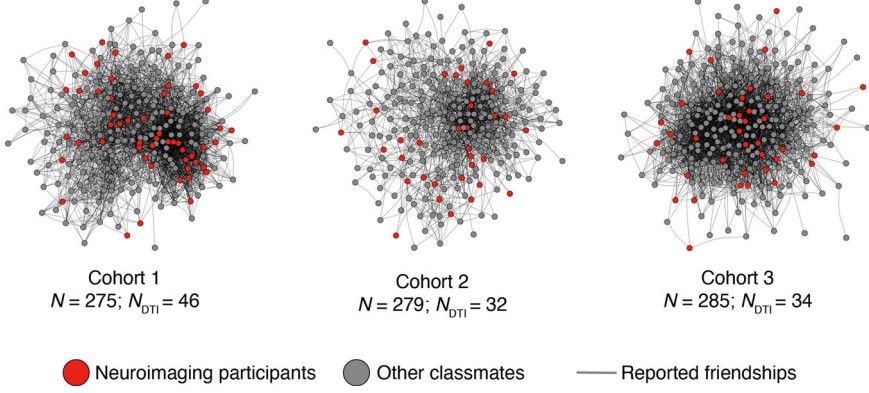

Cohort 1
$N = 275$; $N_{DTI} = 46$

Cohort 2
$N = 279$; $N_{DTI} = 32$

Cohort 3
$N = 285$; $N_{DTI} = 34$

⬤ Neuroimaging participants    ⬤ Other classmates    —— Reported friendships

**Fig. 1 Social network characterization.** Three cohorts of first-year graduate students completed a survey in which they indicated their social ties with other students. These three social networks were reconstructed using this data. Nodes indicate students and lines reflect mutually reported ties between students. Across all three cohorts, a subset of students (red nodes; $N = 112$ after exclusions; see Methods) participated in the dMRI study. The Fruchterman-Reingold layout algorithm, as implemented in the igraph package, was used to position the nodes.

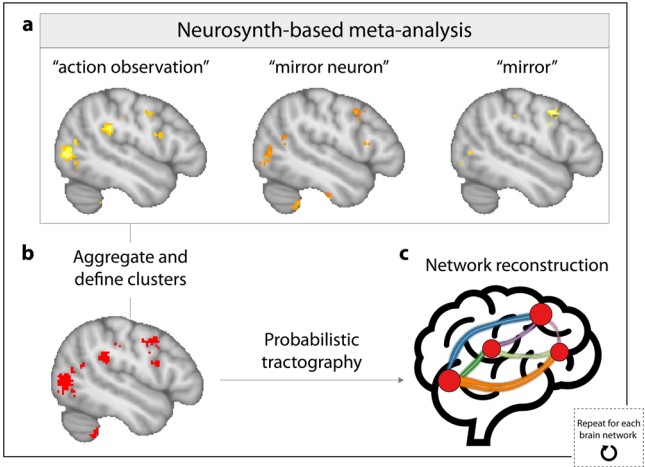

**Fig. 2 Schematic illustrating the process of reconstructing white matter tracts between regions involved in particular facets of socio-affective processing. a** Meta-analysis-based images of brain regions associated with a particular facet of socio-affective processing (e.g., mirroring) were generated by submitting sets of keywords (e.g., action observation, mirror neuron, mirror) to Neurosynth[22]. **b** These images were aggregated across terms in a set, and discrete regions of interest were identified. **c** For each subject, probabilistic tractography was then conducted to trace white matter tracts (colored lines in this schematic image) connecting each pair of brain regions (red nodes). This procedure was repeated to construct the affective processing, mentalizing, and face perception networks (see Methods for further details).

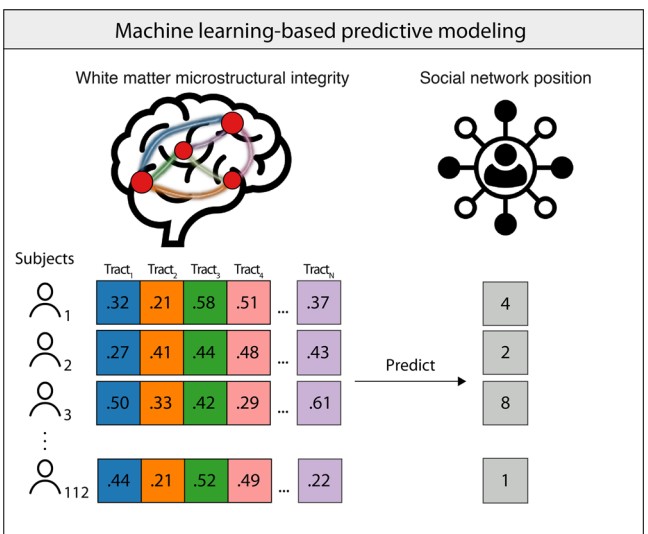

**Fig. 3 Multivariate prediction of social network position characteristics based on patterns of white matter microstructural integrity.** For each subject, average FA was extracted from each white matter tract in a given brain network, and the resulting set of FA values were used as predictors (as shown in different colors) in a ridge regression-based algorithm to predict individuals' social network position characteristics (see Methods for further details). This procedure was performed for the mirroring, affective processing, mentalizing, and face perception brain networks.

details on social network position characteristics). Thus, two measures of brokerage were considered: constraint and betweenness centrality. We examined betweenness centrality to be consistent with prior work that used betweenness centrality as a measure of brokerage in establishing the heritability of this aspect of social network position[12]. We included constraint following our own prior work integrating neuroimaging and social network data[22] and because constraint is a more local measure of brokerage that may be more impacted by individuals' own sociobehavioral tendencies, rather than those of other nearby individuals in the network[23]. In contrast, betweenness centrality captures how often an individual lies on the shortest path between other people in the social network, and thus can be dramatically impacted by factors beyond an individual's own sociobehavioral tendencies[23]. For example, individuals with high betweenness centrality may not function as true brokers, as they may lie on the shortest paths between others merely because they are in close proximity to a true broker[23]. Thus, relative to betweenness centrality, constraint may bear a stronger relationship to the sociobehavioral tendencies that are characteristic of brokers, which in turn are reflected in patterns of white matter microstructural integrity in the brain.

Regions of interest (ROIs) in four a priori defined brain networks associated with different aspects of socio-affective processing (i.e., affective processing, face perception, mentalizing, and mirroring networks) were functionally defined using the meta-analysis tool Neurosynth[24] (Fig. 2). This method provides an approximation of the location of a given ROI if it had been mapped with fMRI in each individual. The three social brain networks examined here were selected based on recent reviews on white matter and social cognition that have emphasized their role in social interactions and processing[20]: Successful social interactions require (1) recognizing and extracting information from others' faces (via regions in the face perception network), (2) quickly understanding their actions, emotions, and intentions through brain regions involved in both

producing and observing actions (via regions in the putative mirroring network), and (3) representing and reasoning about their mental states (via regions of the mentalizing network). Probabilistic tractography was then conducted to trace the white matter tracts connecting every possible pair of ROIs in each brain network (see Methods). Ultimately, the affective processing network consisted of 33 white matter tracts, the face perception network consisted of 50 white matter tracts, the mentalizing network consisted of 41 white matter tracts, and the mirroring network consisted of 49 white matter tracts. To characterize the microstructural integrity of each white matter tract, average fractional anisotropy (FA) values were extracted such that each white matter tract had a corresponding single FA value. FA captures the directional coherence of water in white matter tracts and has been shown to be highly sensitive to factors such as myelination, axonal packing density, and axonal diameter[25].

Using a leave-one-subject-out cross-validation scheme, we used a machine learning algorithm based on ridge regression (see Methods) to test whether patterns of microstructural integrity of white matter tracts within each of the four aforementioned brain networks were predictive of out-degree centrality, in-degree centrality, eigenvector centrality, betweenness centrality, and constraint (Fig. 3). Model performances were measured using the correlation between actual and predicted values of social network position characteristics, and p-values were corrected for multiple comparisons using false discovery rate (FDR) thresholding (see Methods).

**Results: Patterns of white matter microstructure within brain networks involved in social and affective processing are predictive of social network position characteristics.** Patterns of microstructural integrity across white matter tracts in the affective processing network significantly predicted individuals' constraint ($r = 0.263$, $p = 0.002$, $p_{FDR-corrected} = 0.010$), betweenness centrality ($r = 0.240$, $p = 0.006$, $p_{FDR-corrected} = 0.015$), and eigenvector centrality ($r = 0.211$, $p = 0.013$, $p_{FDR-corrected} = 0.026$). Patterns of microstructural integrity across white matter tracts in

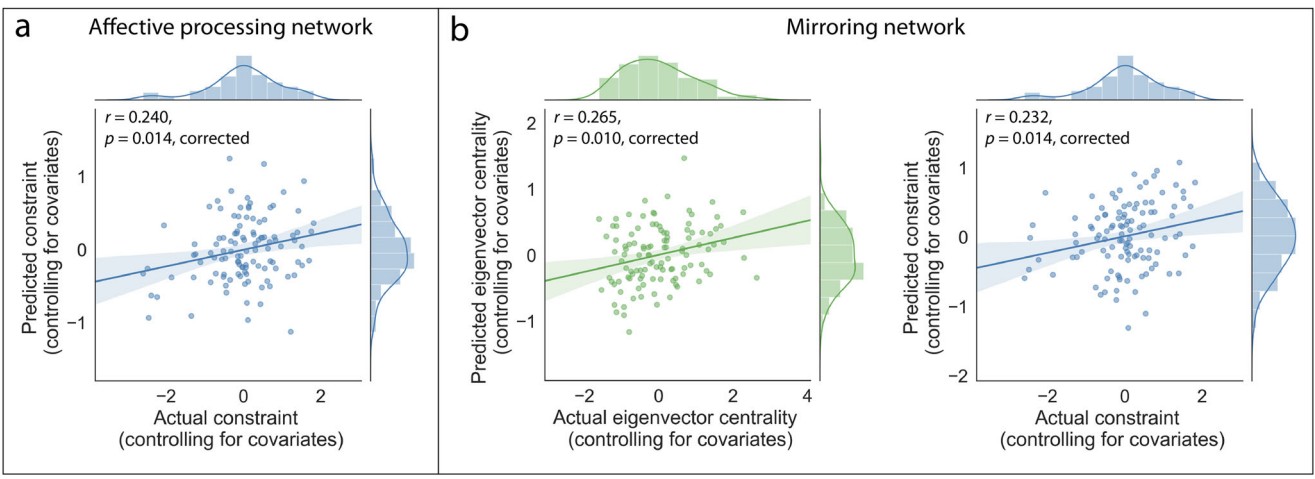

**Fig. 4 Multivariate patterns of white matter microstructural integrity predicted real-world social network position characteristics. a** Whereas patterns of white matter microstructural integrity in the affective processing network were predictive of constraint, **b** patterns of white matter microstructural integrity in the mirroring network were predictive of constraint and eigenvector centrality. For each model, the predictive performance was measured using the Pearson correlation between the actual and predicted values of the social network variable of interest ($N$=112); $p$-values are FDR-corrected; shaded regions indicate 95% CIs. Social network position values were normalized within cohort and square-root transformed prior to analysis. These analyses control for extraversion, demographic characteristics (i.e., age, gender), handedness, and academic cohort.

the mirroring network significantly predicted individuals' constraint ($r = 0.210$, $p = 0.013$, $p_{FDR-corrected} = 0.026$), eigenvector centrality ($r = 0.244$, $p = 0.005$, $p_{FDR-corrected} = 0.019$), and out-degree centrality ($r = 0.239$, $p = 0.006$, $p_{FDR-corrected} = 0.022$). Patterns of microstructural integrity across white matter tracts in the mentalizing network significantly predicted individuals' eigenvector centrality ($r = 0.186$, $p = 0.025$, $p_{FDR-corrected} = 0.033$) and betweenness centrality ($r = 0.172$, $p = 0.034$, $p_{FDR-corrected} = 0.046$). Patterns of microstructural integrity across white matter tracts in the face perception network significantly predicted individuals' betweenness centrality ($r = 0.229$, $p = 0.008$, $p_{FDR-corrected} = 0.015$).

We repeated the above analytic procedure in our primary analyses to test if patterns of microstructural integrity distributed across white matter tracts in each brain network were significantly predictive of social network position characteristics while controlling for demographic variables (i.e., age, gender), as well as handedness and academic cohort. We also controlled for self-reported extraversion, given that extraversion has been associated with social network position characteristics, such as eigenvector centrality[10] (associations between control variables and social network characteristics are reported in Supplementary Note 1). Patterns of microstructural integrity across white matter tracts in the affective processing network were significantly predictive of constraint ($r = 0.240$, $p = 0.005$, $p_{FDR-corrected} = 0.014$) while controlling for these variables. Furthermore, patterns of microstructural integrity across white matter tracts in the mirroring network were significantly predictive of eigenvector centrality ($r = 0.265$, $p = 0.002$, $p_{FDR-corrected} = 0.010$) and constraint ($r = 0.232$, $p = 0.007$, $p_{FDR-corrected} = 0.014$) when controlling for these variables (Fig. 4).

**Results: Patterns of white matter structure across major white matter tracts are predictive of social network position characteristics**. In an exploratory analysis, we used Freesurfer's TRA-CULA (TRActs Constrained by UnderLying Anatomy) tool[26], an algorithm for automated global probabilistic tractography, to reconstruct 18 major white matter tracts for each subject. We then used the same analytic procedure to test if patterns of micro-structural integrity across these major, well-established white matter

tracts can predict social network position characteristics. We observed results similar to those in the primary analysis (see Supplementary Note 2), such that patterns of microstructural integrity were predictive of eigenvector centrality and betweenness centrality, and were also predictive of in-degree centrality and eigenvector centrality when controlling for demographic characteristics (age, gender), extraversion, handedness, and cohort. These results corroborate our primary findings that distributed patterns of white matter microstructural integrity are predictive of social network position characteristics. They also demonstrate that these findings are robust to the use of markedly different data analytic procedures.

**Results: No single white matter tracts are necessary to predict social network position characteristics**. Our primary results suggest that patterns of microstructural integrity distributed across white matter tracts in the mirroring network are predictive of eigenvector centrality, and that patterns of microstructural integrity distributed across white matter tracts in the affective processing and mirroring networks are predictive of constraint, when controlling for covariates. We then sought to conduct exploratory analyses to investigate whether certain white matter tracts were disproportionately contributing to the predictive performance of these models. To this end, we tested whether the exclusion of any single white matter tract would significantly diminish the predictive performance of each full model (i.e., using all $P$ predictors, where $P$ is the number of tracts in a given brain network). We first calculated the true difference in predictive performance between the full model and a model leaving one tract out (i.e., using $P$ - 1 predictors). This true difference value was then compared against a null distribution of 1,000 difference values in predictive performance generated by permutation test-ing. This procedure was repeated $P$ times (i.e., for each tract) for each of these three models, such that the relative contribution of each predictor to each model's predictive performance was evaluated (see Methods).

There were no single tracts that, when omitted, significantly compromised the models' ability to predict eigenvector centrality or constraint from patterns of white matter microstructural integrity in the mirroring network. A similar pattern of null results was observed when testing if the omission of single tracts compromised

the models' ability to predict constraint from patterns of white matter microstructural integrity in the affective network. These results suggest that the exclusion of single white matter tracts from the set of predictors in the affective processing or mirroring networks does not significantly diminish the respective full models' performance in predicting social network position characteristics. Thus, within the brain networks in which patterns of white matter microstructural integrity were significantly predictive of social network position characteristics, no single tract was *necessary* for making such predictions. This procedure was also repeated using the set of predictors derived from the TRACULA analysis, and this analysis also returned null results.

**Results: Single white matter tracts alone are not sufficient to predict social network position characteristics**. We next sought to test whether any single white matter tract in the affective processing, mirroring, face perception, or mentalizing networks would be *sufficient* to predict social network position characteristics, while controlling for extraversion and our other control variables (e.g., demographic variables). For each of the 173 white matter tracts, we used ordinary least squares regression to test whether its microstructural integrity was predictive of any of the five social network position characteristics (out-degree centrality, in-degree centrality, eigenvector centrality, betweenness centrality, and constraint; see Methods). The microstructural integrity of single tracts was not predictive of any of the five social network position characteristics, even when a relaxed threshold for determining significance was used (see Methods). This procedure was also repeated using the set of predictors derived from the TRACULA analysis, and this analysis also returned null results.

The results of our primary analysis demonstrate that distributed patterns of white matter microstructural integrity across tracts in brain networks supporting social and affective processes are predictive of structural characteristics of people's positions in their real-world social networks. Specifically, patterns of white matter microstructural integrity amongst white matter tracts between brain regions associated with affective processing and mirroring were predictive of the extent to which individuals connect otherwise unconnected people and the extent to which individuals are well-connected to other well-connected people in their real-world social networks, above and beyond the effects of demographic variables and extraversion (Fig. 4).

**Patterns of white matter microstructure may shape sociobehavioral tendencies linked to social network position characteristics**. These findings expand on past work demonstrating that various social network position characteristics are heritable individual difference variables that are stable across contexts[12,27]. The genetic basis of social network position characteristics may operate in part via individuals' passive characteristics, which influence how others behave toward them (e.g., their appearance). Consistent with this possibility, physical attractiveness has been shown to be predictive of social status, popularity, and social acceptance[28–31], and people can somewhat accurately infer aspects of strangers' social network position characteristics (i.e., in-degree centrality and constraint) based on their physical appearance[32]. On the other hand, the genetic basis of social network position characteristics may also manifest through active characteristics–e.g., sociobehavioral tendencies that facilitate the occupation of certain kinds of social network positions[13]. For example, such active characteristics might include an individual's sociability, their tendency to introduce their friends to one another, the extent to which they express empathy toward others, their propensity to engage in behavioral mimicry in social

interactions, or some combination of these factors. Individual differences in such sociobehavioral tendencies are likely driven by individual differences in brain structure, but little is known about the relationship between neuroanatomy and social network position characteristics.

Past research has demonstrated that out-degree centrality is linked to individual differences in structural properties of particular brain regions, as well as white matter connectivity[33–37]. The studies referred to here linked individual differences in neural predictors to social network size, which was measured in a variety of ways. However, these different measures of social network size all correspond to out-degree centrality (or in studies using undirected networks, degree centrality). Thus, we use the more precise term out-degree centrality here (since the term network size would imply something different, for example, in sociocentric networks, such as those characterized here, than in egocentric networks). To characterize out-degree centrality, these studies used an egocentric network approach, in which participants enumerate their contacts via free recall or the number of friends that participants have in online communities. This work has yielded important insights into the relationship between brain structure and sociality. At the same time, such approaches have important limitations. For example, it is difficult to disentangle variability in out-degree centrality (self-reported network size) from individual differences in social perception or memory (when self-report is used) or from individual differences in engagement with a particular online platform (when number of friends on social media websites is used)[23,25,38–42]. In contrast, the sociocentric network approach used here incorporates data on social ties provided by each individual in the network and can be used to calculate characteristics of individuals' social network position that take into account broader patterns of social ties (e.g., third party relationships). Thus, the sociocentric network approach can complement the egocentric network approach by capturing a more complete picture of a social network, thereby expanding the types of inferences that can be drawn about people's relative social network position characteristics.

Research in sociology and ecology has demonstrated that social network position characteristics whose calculation often depends on sociocentric network data (e.g., in-degree centrality, eigenvector centrality, constraint, betweenness centrality) have particularly impactful consequences in real-world social networks. These include measures of evolutionary fitness and likelihood of survival across a variety of social species[43,44], as well as social influence[2], professional success[1,2,9], others' perceptions of one's competence and leadership[5,45], and the likelihood of becoming the target of negative gossip and scapegoating[46]. Furthermore, whereas out-degree centrality has not been found to be heritable, other, often sociocentrically-derived, social network position characteristics have been shown to be heritable individual difference variables[12]. Thus, the latter may constitute stable traits that are relatively invariant across contexts[27]. Indeed, a growing body of research has integrated sociocentric network analysis and neuroimaging to demonstrate that people spontaneously encode and track the extent to which others hold positions of in-degree centrality[47,48], eigenvector centrality, and brokerage[22] in real-world social networks. These findings suggest that individuals spontaneously retrieve complex knowledge about people's relative social network position characteristics that may be crucial for informing cognition and behavior. However, the neural predictors and sociobehavioral tendencies associated with such social network position characteristics are not well understood in social neuroscience and psychology, given that the data necessary to calculate these characteristics (i.e., sociocentric network data) is seldom collected.

**Patterns of white matter microstructural integrity in brain networks supporting socio-affective processing predict social network position characteristics**. The localization of the current results can shed light on the types of active characteristics, or sociobehavioral tendencies, that may be associated with particular social network position characteristics. In particular, patterns of microstructural integrity of white matter tracts in the affective processing and mirroring networks were predictive of constraint (an inverse measure of brokerage), beyond the effects of demographics (age, gender), handedness, cohort, and extraversion. Brokers connect people who would not otherwise be connected and thus wield leverage in controlling the flow of resources (e.g., information) and in coordinating behavior across local social ties[49]. Given that occupying positions of brokerage involves interacting with different groups of people, brokers may be exceptionally skilled in adapting their thoughts and behavior to meet the variable demands of their diverse social environment. Indeed, past work has shown that across different contexts, people occupying positions of brokerage are characteristically high in self-monitoring[50–53], which is associated with an intuitive sensitivity to subtle social cues and with the ability to modify one's behavior to adapt to social circumstances[54,55]. Individuals high in self-monitoring have been shown to closely monitor the thoughts, actions, and feelings of people around them[56,57] and also invest considerable effort in providing emotional help[58] and advice[59] to their contacts. Such sociobehavioral tendencies may be driven by individual differences in patterns of white matter microstructural integrity across the affective processing and mirroring networks. The affective processing network may support the monitoring and interpretation of emotions and the regulation of one's own emotions[60–62], and the mirroring network may mediate the representation, understanding, and mimicry of the actions of others[63–65]. Given that brokers are highly attuned to cues of situational appropriateness, they are likely exceptionally skilled at accurately perceiving and interpreting the emotions and actions of others. Brokers may also be particularly likely to exhibit social chameleon-like behavior such that they engage in nonconscious mimicry and imitate the behaviors of their social contacts. Such behavior has been shown to be associated with increased mutual feelings of affiliation, rapport, and liking[66] and would be conducive to bridging disparate groups of people that may behave in different ways.

Additionally, patterns of microstructural integrity of white matter tracts in the mirroring network were predictive of eigenvector centrality, a prestige-based measure of centrality that takes into account not only an individual's own centrality but also the centralities of their contacts[67]. Given the relative dearth of previous research investigating cognitive and behavioral traits associated with eigenvector centrality, the link between eigenvector centrality and the cognitive and behavioral tendencies associated with the mirroring network is unclear. However, being highly attuned to social cues and having the ability to effectively understand and imitate the actions of others may be particularly characteristic of individuals occupying positions of high eigenvector centrality, as such tendencies are conducive to being well-liked[66] and may also lead to the formation and maintenance of social ties with other well-connected individuals.

Patterns of white matter microstructure were not significantly predictive of out-degree centrality, in-degree centrality, and betweenness centrality beyond the effects of covariates. While it is difficult to interpret null findings, we note that this may be attributable to a variety of factors. For example, given that a sociocentric network approach was used here, participants only enumerated contacts within their bounded social network; thus, the measure of out-degree centrality used here was limited to capturing nominations of friends within the social network.

While the bounded social networks characterized here were composed of people in a rural, isolated location who largely live, eat, socialize, and study with one another (see Methods), participants may have had friends outside of their academic cohorts that were not characterized. It is possible that patterns of white matter microstructure in social processing networks would be predictive of out-degree centrality if friendships with people outside of participants' academic cohorts were taken into account. Alternatively, more targeted analyses in tracts defined a priori may yield significant predictions of out-degree centrality, whereas the current study corrected for multiple comparisons across multiple sets of analyses.

Patterns of white matter microstructural integrity were also not predictive of in-degree centrality, but they were significantly predictive of eigenvector centrality. This suggests that individual differences in patterns of white matter microstructural integrity are related to individual differences in characteristics of the social network position that take into account indirect relationships and broader patterns of social ties (i.e., being well-connected to well-connected others) that go beyond more local characteristics (i.e., the number of direct nominations one receives from others).

At the same time, consistent with the possibility that constraint, as a more local measure of brokerage, is more strongly related to individuals' sociobehavioral tendencies (in contrast to betweenness centrality, a measure of brokerage that can be impacted by more distal factors, such as lying on the shortest path between others due to being close to a true broker[41]), patterns of white matter microstructural integrity were not predictive of betweenness centrality, but were predictive of constraint.

**Patterns of white matter microstructural integrity in the mentalizing network did not predict social network position characteristics.** While patterns of white matter microstructural integrity within the affective processing and mirroring networks were predictive of social network position characteristics when controlling for covariates, those in the mentalizing network were not. Thus, structural characteristics of individuals' positions in their social networks appear to be linked to connectivity in brain networks involved in rapid, automatic processes involved in understanding and relating to others and their emotional states (e.g., mirroring and affective processing), but not in those supporting more cognitive facets of interpersonal understanding. Understanding cues to the internal states of others – i.e., the construct of empathy – may be broken down into emotional and cognitive empathy, which are distinct processes with distinct neural mechanisms[68–72]. On the one hand, regions in the putative human mirror neuron network support emotional empathy, which is an automatic, rapid process that mediates one's emotional, sensorimotor, and visceral response to the affective state of another person. On the other hand, a different set of brain regions supports cognitive empathy, which is a comparatively slow, effortful process that mediates one's conscious ability to understand or explicitly recognize the mental states (e.g., perspectives, intentions) of others[70,71,73,74].

Here, "empathy" was one of the terms used to define brain regions in the mentalizing network (see Methods). Whereas this term may have yielded brain regions associated with *both* cognitive and emotional empathy, terms such as "mirror neuron" and "mirror" that were used to define regions in the mirroring network are aligned with emotional, but not cognitive, empathy. Thus, it is possible that individual differences in patterns of white matter microstructural integrity in the mirroring network reflect individual differences in emotional empathy (and not cognitive empathy) and that these individual differences are linked to

individuals' social network position characteristics. Indeed, past work has shown that individual differences in the microstructural integrity of white matter tracts connecting perception and action-related regions (i.e., regions in the mirroring network) and regions involved in affective processing are predictive of emotional, but not cognitive, empathy[62]. In contrast, individual differences in patterns of white matter microstructural integrity in the mentalizing network likely at least partially reflect individual differences in cognitive empathy. It is possible that cognitive empathy is not linked to people's social network position characteristics. It is also possible that cognitive empathy relies on more domain-general neural mechanisms[75,76], whereas emotional empathy relies on more domain-specific neural circuitry[77], which could make associations between cognitive empathy and its structural correlates less robust than those between emotional empathy and its structural correlates. Furthermore, a limitation of the current study is that a group-level meta-analytic map was used to define regions of interest associated with the mentalizing network. However, recent work has demonstrated that the spatial specificity of the mentalizing network is highly heterogeneous across individuals[78]. This finding may explain the current study's null results when using patterns of white matter microstructural integrity within the mentalizing network to predict social network position characteristics, and future work may benefit from using functional localizers to identify mentalizing regions of interest that would be used as seed regions in probabilistic tractography.

**Conclusions and future directions**. We suggest that future work build on the current findings by examining the extent to which individual differences in the processes supported by the aforementioned brain networks (e.g., emotional empathy, facets of affective processing, such as empathic accuracy) mediate the relationship between brain structure and social network position characteristics. Future work should also examine the extent to which individual differences in brain structure precede or result from individual differences in social network position characteristics. It is possible that brokers attain their advantageous social network position characteristics through distinctive capacities for interpersonal understanding, which are reflected in structural characteristics of brain networks involved in mirroring and affective processing. It is also possible that occupying a high-brokerage position in one's social network places more demands on one's capacity for social and affective information processing (e.g., due to the need to be sensitive to differing social and emotional cues in different social groups and to flexibly modulate one's behavior to suit different contexts). Over time, this may lead to structural differences in the brain networks that support such processing.

In addition, while the current study pooled data from three different bounded communities, all participants were graduate students at a university in the United States. Future work could extend these findings by examining the relationship between white matter microstructure and social network position characteristics in different cultural settings and at different stages of development. Furthermore, given a large enough sample size, future work would also benefit from training and testing models on different samples of individuals, as this would shed light on whether relationships between white matter microstructural integrity and social network position characteristics are consistent across contexts and communities. Additionally, while the current study used FA as a measure of microstructural integrity, future work may benefit from testing if other measures of white matter microstructure, such as axial diffusivity, mean diffusivity, and radial diffusivity, or the number of probabilistic tractography

streamlines, are predictive of social network position characteristics.

Future work may also benefit from using functional localizers to identify seed regions of interest, as the location of functionally defined brain regions may vary from person to person[79]. In particular, aspects of the mentalizing network have recently been shown to vary substantially across individuals[78]. Thus, using functional localizers to define subject-specific seed regions of interest would likely confer greater sensitivity for detecting systematic links between patterns of white matter microstructural integrity and traits such as social network position characteristics. Furthermore, with functionally localized subject-specific ROIs, it is also possible to test if the size of ROIs may systematically affect FA-based prediction of social network position characteristics.

Additionally, the current work focused on patterns of white matter microstructure across tracts *within* brain networks, as it was motivated by an interest in linking individuals' social network position characteristics to anatomical connectivity among brain regions involved in particular mental functions (e.g., linking characteristics of tracts connecting brain regions involved in affective processing to measures of social network centrality). We did not test whether patterns of white matter microstructure across functionally defined tracts *between* brain networks were predictive of social network position characteristics. It is conceivable that patterns of white matter microstructure across between-network tracts and, correspondingly, interactions between different mental functions (to the extent that tracts between such networks support interactions between the corresponding mental functions), also play an important role in shaping social behavioral tendencies and therefore an individual's social network position characteristics. We did, however, find that patterns of white matter microstructure across TRACULA-defined major white matter tracts were predictive of social network position characteristics. Future work could benefit from specifically considering tracts between different functionally defined brain networks.

Our exploratory analyses indicated that no single white matter tract in particular was necessary for predicting social network position characteristics. Furthermore, no single white matter tract was sufficient for predicting social network position characteristics on its own. Rather, our results suggest that multivariate patterns of microstructural integrity values derived from sets of white matter tracts were necessary for predicting social network position characteristics. These findings expand on recent neuroimaging research demonstrating the utility of data-driven machine learning models in mapping relationships between multivariate sets of neural predictors and person-level outcomes. Such data-driven predictive modeling frameworks have been adopted to link functional neuroimaging data to a wide range of social, cognitive, and behavioral traits[80–85]. In particular, recent work has demonstrated that whole-brain patterns of resting-state functional connectivity are predictive of one's location in a real-world social network[86]. Taken together with the current results, these findings suggest that applying machine learning to high-dimensional neuroimaging data is a fruitful approach for gaining insight into how brain structure and function relate to individuals' positions in their real-world social networks.

## Methods

**Social network characterization**. Subjects who completed the social network survey were from three different cohorts of first-year students in a graduate program at a private university in the United States who participated as part of their coursework on leadership. The total size of all three cohorts was 842 students, and 839 students participated in the social network survey, resulting in an overall response rate of 99.6% ($N_{Cohort-1}$ = 275, 91 females, response rate = 99.3%; $N_{Cohort-2}$ = 279, 89 females, response rate = 100%; $N_{Cohort-3}$ = 285, 120 females, response rate = 99.7%). For each cohort, an online social network survey was administered 3-4 months after

the subjects had arrived on campus. Subjects followed an e-mailed link to the study website where they responded to a survey designed to assess their position in the social network of students in their cohort of the academic program. The survey was adapted from prior work[1,22,87,88]. It read, "*Consider the people with whom you like to spend your free time. Since you arrived at [institution name], who are the classmates you have been with most often for informal social activities, such as going out to lunch, dinner, drinks, films, visiting one another's homes, and so on?*" A roster-based name generator was used to avoid inadequate or biased recall. Subjects indicated the presence of a social tie with an individual by placing a checkmark next to their name. Subjects could indicate any number of social ties and were not constrained by a time limit. The bounded social networks characterized here were composed of people in a rural, isolated location who predominantly lived, ate, socialized, and studied with one another. The social network survey used here inquired only about students' interactions with other members of their academic cohort. Thus, the current approach does not capture the students' social ties that exist outside of their cohort of classmates (e.g., relationships with family members, friends outside of the program). That being said, the current study was conducted at a relatively insular and remotely located institution where subjects' contacts outside of campus likely play a smaller role in their daily lives relative to their everyday, in-person interactions with their classmates. All data collection procedures were performed in accordance with the standards of the Dartmouth College Institutional Review Board.

Each cohort's social network data was analyzed using igraph in R[89]. The social networks of the three cohorts are depicted in Fig. 1. Five social-network-derived metrics were calculated for each subject who participated in the neuroimaging study: out-degree centrality, in-degree centrality, eigenvector centrality, betweenness centrality, and constraint. An unweighted graph was used to calculate each of these social network position characteristics for each subject, as described in greater detail below. For descriptive purposes, for each cohort's social network, we then calculated the mean and median numbers of social ties across individuals (i.e., average total degree centrality, summing incoming and outgoing ties for each individual) and the reciprocity of the graph, which refers to the probability that person $i$ nominated person $j$ as a friend if person $j$ nominated person $i$ as a friend (mean social ties$_{Cohort-1}$ = 91, median social ties$_{Cohort-1}$ = 77, reciprocity$_{Cohort-1}$ = 0.53; mean social ties$_{Cohort-1}$ = 78, median social ties$_{Cohort-1}$ = 70, reciprocity$_{Cohort-1}$ = 0.49; mean social ties$_{Cohort-1}$ = 55, median social ties$_{Cohort-1}$ = 46, reciprocity$_{Cohort-1}$ = 0.48).

### Social network position characteristics

*Out-degree centrality.* The out-degree centrality of an individual was calculated as the sum of the individual's outgoing social ties (i.e., the number of people whom the individual nominated).

*In-degree centrality.* The in-degree centrality of individual was calculated as the sum of individual's incoming social ties (i.e., the number of times the individual was nominated by others).

*Eigenvector centrality.* A graph consisting of nodes connected by edges can be characterized by an adjacency matrix $\mathbf{A}$, populated by elements such that $a_{ij} = 1$ if nodes $i$ and $j$ are directly connected, and $a_{ij} = 0$ if these nodes are not connected. The eigenvector centrality of each node is given by the eigenvector of $\mathbf{A}$ in which all elements are positive. The requirement that all elements of the eigenvector must be positive yields a unique eigenvector solution (that is, that corresponding to the greatest eigenvalue). Here, when computing eigenvector centrality, the directionality of the graph was preserved; in the event of asymmetric relationships, only incoming, rather than outgoing, ties were used to compute eigenvector centrality.

*Betweenness centrality.* The betweenness centrality of an individual was calculated as the proportion of shortest paths between two given nodes that pass through the individual. An unweighted, undirected graph was used to estimate betweenness centrality.

*Constraint.* The constraint of actor $i$ is given by the following equation, where $P_{ij}$ corresponds to the proportion of $i$'s direct social ties accounted for by their tie to actor $j$. The inner summation approximates the indirect constraint imposed on $i$ by other actors, $q$, who are socially connected to both $i$ and $j$ (mutual friends of $i$ and $j$):

$$\text{Constraint}_i = \sum_{j=1}^{n} \left( P_{ij} + \sum_{q=1}^{n} P_{iq} P_{qj} \right)^2 \qquad (1)$$

An unweighted, undirected graph was used to estimate constraint; that is, the presence of any social tie, irrespective of its direction or if it was reciprocated, was used to compute the constraint of each node. Constraint was then negated to yield a measure of network brokerage.

All social network position characteristic data were normalized (z-scored) within-cohort. These data were then concatenated across all three cohorts. For each of the social network position characteristics, we then applied the square-root transformation, given that the distributions for each of the social network position characteristics were positively skewed. Correlations between social network characteristics are reported in Supplementary Table 1.

**Neuroimaging subjects.** A subset of 130 individuals who had completed the social network survey completed a subsequent dMRI study ($n_{Cohort-1}$ = 54; $n_{Cohort-2}$ = 36; $n_{Cohort-3}$ = 40). Subjects in cohort 1 were scanned 6-7 months after their arrival on campus, subjects in cohort 2 were scanned 7-8 months after their arrival on campus, and subjects in cohort 3 were scanned within the first month of their arrival on campus. Subjects provided informed consent in accordance with the policies of the institution's ethical review board. Of the 130 subjects, data from 18 subjects were excluded from analysis due to excess movement. Data from the resulting 112 subjects (40 female) aged 24–35 ($M$ = 27.78, $SD$ = 2.01) were used for analysis. The neuroimaging study was advertised to all students in each cohort via email. All students who were interested in participating and who passed a standard MRI safety screening participated in the DTI scan.

**dMRI acquisition.** Magnetic resonance imaging was conducted with a Philips Achieva 3.0 Tesla scanner using a 32-channel phased array head coil. Diffusion-weighted images were collected using 70 contiguous 2 mm thick axial slices with 32 diffusion directions (91 ms TE, 8845 TR, 1000 s/mm² b-value, 240 mm FOV, 90° flip angle, 1.875 mm × 1.875 mm × 2 mm voxel size). Thirty-three diffusion-weighted volumes were collected per subject. High-resolution anatomical images were also acquired using a T1-weighted MPRAGE protocol (8.2 s TR; 3.7 ms TE; 240 × 187 FOV; 0.938 mm × 0.938 mm × 1.0 mm).

**Diffusion tensor imaging.** We performed standard preprocessing steps using the Diffusion toolbox in FSL 5.0.10[90], which included brain extraction, eddy current correction, and motion correction. We then used FSL's *dtifit* to fit a diffusion tensor model at each voxel to generate an FA map for each subject (FA serves as a general marker of white matter microstructural integrity)[25]. We also used FSL's *BEDPOSTX*[90,91] to model crossing fibers and white matter fiber orientations in each voxel. Both linear and non-linear methods were used to align subjects' fractional anisotropy images in native space to MNI152 standard space. FSL FLIRT (12 degrees of freedom, corratio cost function) was used to generate an affine transformation matrix to align fractional anisotropy images to T1 anatomical images. FSL FLIRT was used to generate an affine transformation matrix to align T1 space to MNI152 standard space, and FSL FNIRT used this "affine guess" to generate a non-linear warpfield to align T1 space to MNI152 standard space. FSL FNIRT was then used to apply the first affine FLIRT matrix (i.e., native space to T1 space transform) and the FNIRT warpfield in one step to transform fractional anisotropy images to MNI152 standard space.

**Defining social processing networks: ROI definition and probabilistic tractography.** Probabilistic tractography was conducted to reconstruct white matter tracts between ROIs associated with each of four facets of social processing: affective processing, mirroring, mentalizing, and face perception. This process was used to define the affective processing, mirroring, mentalizing, and face perception networks (i.e., each of the four sets of ROIs, and the tracts connecting its constituent regions). A schematic of this procedure is visualized in Fig. 2.

Keywords were submitted to Neurosynth[24] to generate whole-brain meta-analysis-based images of networks of brain regions involved in affective processing (emotion, valence, affective, mood, arousal), mirroring (action observation, mirror neuron, mirror), mentalizing (theory of mind, mentalizing, empathy), and face perception (faces, face recognition, face). For each brain network, the meta-analysis-based images associated with each keyword were aggregated, and the FSL *cluster* command was used to identify the discrete ROIs in each of the aggregated meta-analysis images. The FSL *FLIRT* and *FNIRT* commands were used to transform ROIs from standard space to each subject's native diffusion space. This method provides an approximation of the location of a given ROI if it had been mapped with fMRI in each individual. Each ROI was then dilated by a single voxel and was masked using a brain mask to create more liberal ROI masks to ensure that ROIs would extend into neighboring white matter. Due to being exceptionally large and/or spanning multiple regions, a small subset of ROIs were then masked using subject-specific anatomical masks generated using the Freesurfer anatomical parcellation algorithm[92–94] in order to split the ROI into smaller constituent ROIs, which were subsequently used for analysis. Details regarding the size of each ROI, the anatomical regions associated with each ROI, and whether the ROI was masked using subject-specific anatomical masks are provided in Supplementary Tables 2–5.

For each brain network, the FSL *probtrackx2* command was then used to perform probabilistic tractography between every possible pair of ROIs within each of the left and right hemispheres (i.e., tractography was not performed to trace inter-hemispheric tracts), with a contralateral hemisphere exclusion mask and a brainstem exclusion mask. Two-mask seeding was used, and 1,000 probabilistic tract streamlines were taken at each voxel within each mask, which allows resulting tractography maps to include streamlines originating from and terminating in each ROI. For each proposed white matter tract, if more than half of the subjects yielded zero fiber tracts (i.e., if more than half of the subjects did not have a valid tracing), the corresponding white matter tract was excluded from further analysis. The remaining output connectivity distribution maps were divided by the corresponding total number of existing streamlines to normalize and convert the images into probabilistic maps. These probabilistic maps were then thresholded such that all voxels with a probability below 1% were zeroed in order to reduce

false-positive fiber tracts. The resulting probabilistic maps were then binarized within each subject and summed across all subjects in standard space to create group-level tractography images. The group-level tractography images were manually inspected to determine thresholds that minimized spurious connections. Tracts that were identified as spurious upon manual inspection were excluded from further analysis. The resulting group-level images were then transformed back into each subject's native diffusion space. For each subject, we then masked the resulting images using subject-specific white matter tissue masks (generated by FSL), conservatively thresholded at 50% in order to eliminate spurious white matter, and binarized. For each subject, FA values were extracted from each voxel from each white matter tract image using subject-specific FA maps thresholded at 0.2, which is a standard practice to create conservative FA maps. The resulting FA values were averaged across voxels within each tract to yield a single FA value reflecting the microstructural integrity of a given white matter tract.

**Structural connectome-based predictive modeling of social network position characteristics**. We tested if inter-individual variability in patterns of white matter microstructural integrity would be predictive of individuals' positions in their real-world social network (Fig. 3). That is, the independent variables consisted of a multivariate set of predictors measuring white matter microstructural integrity, and the outcome variable consisted of a given social network position characteristic. We used Scikit-learn[95] to implement the predictive modeling analysis. Using Scikit-learn's Pipeline function, we created an algorithm that performed two steps in sequence on the training data for each fold (models fit to each fold's training data were used to predict social network position characteristics based on white matter microstructural integrity values in the corresponding testing data): (1) normalize the predictors using Scikit-learn's StandardScaler function (which subtracts the mean and scales to unit variance) and (2) implement ridge regression. Given the multicollinearity among the predictors, regularized ridge regression was used. We used a nested cross-validation scheme to perform hyperparameter tuning using a grid search procedure (i.e., optimizing the lambda ($\lambda$) regularization hyperparameter from a grid/range of values logarithmically spaced between $10^{-5}$ and 10), such that the training data of each of the 10 outer data folds was further subdivided into 10 inner folds consisting of sub-training and validation datasets. Within each of these inner folds, for each hyperparameter value provided in the hyperparameter grid, the algorithm was trained on the sub-training data and tested on the validation data. The hyperparameter value used in the model with the best performance across all validation sets was identified as the optimal hyperparameter for the corresponding outer training fold. Using this optimal hyperparameter, the algorithm was trained on the outer fold's training data and tested on the outer fold's testing data. This process was repeated independently for each of the ten outer data folds. This procedure yielded a predicted social network position characteristic value for each subject in the sample. Out-of-sample performance was evaluated by calculating the Pearson $r$-value between predicted and actual social network position characteristic values. All reported $p$-values reflect one-sided tests of if participants' actual social network position characteristics were positively associated with the social network position characteristics predicted by the trained models, based on participants' dMRI data (negative associations would not be interpreted in this context). Given that we tested if patterns of white matter microstructural integrity associated with brain networks were predictive of five different social network position characteristics, we corrected for multiple comparisons across these five sets of statistical tests using FDR thresholding.

To test whether inter-individual variability in patterns of white matter microstructural integrity would be predictive of social network position characteristics above and beyond the effects of demographics and extraversion, we repeated the analysis described above while controlling for age, gender, handedness, cohort, and self-reported extraversion. Extraversion was assessed using the relevant items of the Big Five 44-item inventory[96].

**Testing whether any single white matter tracts were necessary for predicting social network position characteristics**. The primary results demonstrate that patterns of microstructural integrity distributed across white matter tracts in the affective processing network are predictive of constraint, and that patterns of microstructural integrity distributed across white matter tracts in the mirroring network are predictive of constraint and eigenvector centrality, even while controlling for variables including age, gender, handedness, cohort, and extraversion. Thus, in these three models (two models predicting constraint; one model predicting eigenvector centrality), we tested if any predictors (i.e., where a predictor corresponds to the microstructural integrity of a given white matter tract) in particular were disproportionately contributing to significant predictions of social network position characteristics.

For each of the three models mentioned above, the following procedure was implemented. We created $P$ additional 'leave-one-tract-out' models where $P$ reflects the number of predictors (tracts) in the corresponding full model. Each of these additional models excluded one of the $P$ predictors that had been included in the corresponding full model such that the number of predictors in the resulting leave-one-tract-out model was equal to $P$ - 1. The permutation testing procedure used in the primary analysis was then used to calculate the performance of each of the $P$ leave-one-tract-out models in predicting the relevant social network position characteristic. For each of the $P$ leave-one-tract-out models, the difference in the model's predictive performance (i.e., the $r$-value measuring the correlation between

predicted and actual values) and that of the full model was calculated. We then tested the statistical significance of this difference in predictive performance (i.e., tested if excluding a given predictor significantly diminished the full model's performance). To this end, we used a permutation testing procedure where social network position characteristics were randomly shuffled across subjects in each of 1,000 permuted datasets. Here, in each permuted dataset, $r$-values were obtained for the full model and for each of the $P$ leave-one-tract-out models. For each of the leave-one-tract-out models, we then calculated the difference between its $r$-value and that of the full model within each permuted dataset; this produced a null distribution of 1,000 difference values for each of the leave-one-tract-out models. We then calculated a $p$-value measuring the frequency with which the true difference value was greater than the permuted difference values in this null distribution.

**Testing whether any single white matter tracts alone were sufficient for predicting social network position characteristics**. We next tested if the microstructural integrity of any single white matter tract in the social processing brain networks defined above was alone *sufficient* to predict social network position characteristics while controlling for extraversion and demographics. To do so, we used Scikit-learn[95] to implement ordinary least squares regression in a predictive modeling framework. There were 173 white matter tracts in total across the four examined brain networks. Thus, to test if microstructural integrity of each of these 173 white matter tracts were predictive of the five social network position characteristics (out-degree centrality, in-degree centrality, eigenvector centrality, betweenness centrality, and constraint), we conducted 865 statistical tests.

For each statistical test, we used Scikit-learn's Pipeline function to create an algorithm that performed two steps in sequence on the training data for each fold (models fit to each data fold's training data were used to predict social network position characteristics based on the white matter microstructural integrity value in the corresponding testing data; see "Structural connectome-based predictive modeling of social network position characteristics" for details of the data-folding procedure): (1) normalize the predictors using Scikit-learn's StandardScaler function and (2) implement linear regression. Out-of-sample performance was evaluated by calculating the Pearson $r$-value between predicted and actual social network position characteristic values. All reported $p$-values reflect one-sided tests of if participants' actual social network position characteristics were positively associated with those predicted by each trained single-tract model (negative associations would not be interpreted in this context). To correct for multiple comparisons across all 865 statistical tests, we used FDR thresholding. This procedure corrects for the total number of models tested and corresponding results indicated that data from single tracts could not predict social network position characteristics, unlike the data from distributed patterns of tracts used in our main analyses. At the same time, we note that the threshold used in these exploratory single-tract analyses (correcting for multiple comparisons across 865 statistical tests) was much more conservative than that used in our main analyses, where predictors from each brain network were combined into a single model to predict each examined social network position characteristic. Therefore, as an additional point of comparison, we also examined results of single-tract analyses using a less conservative threshold for determining statistical significance: Given that we tested if microstructural integrity associated with single tracts was predictive of five different social network position characteristics, we corrected for multiple comparisons across these five sets of statistical tests. This relaxed threshold yielded identical results to the results obtained using a more conservative threshold (see Results and Discussion section).

**Statistics and reproducibility**. Neuroimaging data were normalized within cohorts and were then pooled across cohorts to yield a sample size of 112 participants. Statistical analyses were conducted as described in the *Structural connectome-based predictive modeling of social network position characteristics, Testing whether any single white matter tracts were necessary for predicting social network position characteristics*, and *Testing whether any single white matter tracts alone were sufficient for predicting social network position characteristics* sections in *Methods*. As discussed in the *Conclusions and future directions* section, the extent to which the observed relationships between white matter microstructural integrity and social network position characteristics are consistent across contexts and communities is unknown. Future work may shed light on how such relationships vary across contexts and communities.

**Reporting summary**. Further information on research design is available in the Nature Research Reporting Summary linked to this article.

# Data availability
Data used for visualizations in Fig. 4 are available in Supplementary Data 1. All other data that support the findings of this study are available upon reasonable request.

# Code availability
Code that supports the findings of this study is available upon reasonable request.

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

## Acknowledgements
This work was supported by funds from the Dartmouth Brain Imaging Center, a Dartmouth Cross-disciplinary Collaboration Seed Grant, New Faculty Startup Funds from the University of California, Los Angeles, a Graduate Research Mentorship Award, and National Science Foundation Grant No. SBE-2048212. Figure 2 uses an icon that was made by Meghan Hendricks from www.thenounproject.com. Figure 3 uses icons made by Meghan Hendricks and Anconer Design from www.thenounproject.com.

## Author contributions
R.H., C.P., T.W., and A.M.K. designed research; C.P. and A.M.K. performed research; R.H. and J.A.H.C analyzed data with guidance from R.S.C., C.P., and A.M.J.; and R.H., R.S.C., and C.P. wrote the paper with input from all authors.

## Competing interests
The authors declare no competing interests.
