## [Peer Review File · Communications Biology]

Reviewers' comments:

Reviewer #1 (Remarks to the Author):

In this paper, the authors employed diffusion MRI to study brain network features (i.e., white matter connectivity) that contribute to social network characteristics (i.e., individual differences in social network positions). They found distributed patterns of white matter microstructural integrity, especially those supporting social brain networks and major white matter tracts, are predictive of real-world social network position characteristics.

This is an elegantly simple study that has important implications for the 'brain network underlying social network' literature. The study is well-designed and has collected enormous multimodal data (behavioral + DTI); the results are clear, consistent with previous studies; and the conclusions they draw are well-justified from their results. While I enjoy reading this paper very much and recommend publication, but would like to see some changes first (mainly clarifications and discussions).

Major points

1. The authors collected data from three different cohorts. While the distributed white matter microstructure models are very impressive to predict real-world stuff, I am concerned with the generalizability of the models. Is it possible to train a model based on one cohort and then test on another? The theoretical question here is that: do all three cohorts share the same neural mechanism? I know this remark might be too harsh and the sample size in each cohort might not be enough for decent machine learning modelling, but at least the authors should discuss them for future directions.

Minor points

2. Page 4, line 96. A subset of subjects underwent DTI scans. Did author strategically choose those DTI subjects based on their critical network positions (optimal selection) or just randomly collected data from any subjects willing to be scanned. Please clarify

3. Page 4, line 107. The definition and meaning of the network metric 'constraint' is unclear. Does 'constraint' mean the extent to which one connects to unconnected/isolated individuals? The description is a bit of cursory and obscure. In addition, what's the statistical relationship between five social network position characteristics (i.e., out-degree centrality, in-degree centrality, eigenvector centrality, betweenness centrality, and constraint) in the current behavioral data? I know these metrics are conceptually different, but it will be helpful for readers to see how real data speaks. The supplementary material only provides t-test, but no correlations across all subjects.

4. Page 12, line 372. It is still possible that other types of white matter microstructure (e.g., number of streamlines, MD, AD, RD, or even FA in certain segments of a tract) that can predict out-degree centrality, in-degree centrality, and betweenness centrality. The authors might not need to run extra analyses but needs to mention this limitation in the discussion.

5. Page 13, line 414. The way to define mentalizing network using Neurosynth in this paper can be inaccurate, as a recent study suggested that the spatial specificity of mentalizing network is highly heterogeneous across subjects (see Wang et al., 2021 NeuroImage; doi: 10.1016/j.neuroimage.2021.118115). This limitation could be discussed, as it might contribute to the null results on mentalizing network, and future research should be encouraged to use more optimal ways (i.e., functional localizer) to validate the null results.

6. Page 18, line 596. The author could improve the paper by providing more information on the ROI definition. For each social brain network, how many ROIs were selected from Neurosynth maps? What were their anatomical labels? How many voxels in each ROI, and how many ROIs were bilateral or unilateral. Was tractography implemented only among the same hemisphere ROIs or included cross-hemisphere ROIs. A table listing above information will be extremely helpful for

open science.

Reviewer #2 (Remarks to the Author):

Investigating the contribution of white matter connectivity to social behavior is essential to understand the neural basis of our social self in health and disease. This manuscript takes on the extra challenge to investigate the link between white matter connectivity and one's social position in real-world networks, a critical question to understand the link between white matter integrity and real-world behaviors. The study is building upon a beautiful and large dataset of neuroimaging scans from 3 independent cohorts of graduate students, which social networks have been fully sociologically characterized. Building upon previous work (e.g.(Wang and Olson 2018)), the authors studied white matter integrity by considering fractional anisotropy not only within major white matter tracts, but also within fasciculi identified for social cognition. By using machine-learning, they discover that neuroanatomy of white matter parallels one's position within its' social network in terms of eigenvector centrality, and brokerage, two sociological variables of the network. Please find below a detailed description of questions that could be addressed by the authors.

Major questions:

1. My main concern, which could be overcome by providing additional information and analyses, goes towards quantification of relationship in-between social network position characteristics; and how this is impacting on the model. Statistical tests would be needed to investigate correlation between each social network position characteristic (in-degree centrality, out-degree-centrality, eigenvector centrality, constraint, betweenness) with one other. It is unclear from the Methods section (L662) if the social characteristics are tested simultaneously by the model or if they are tested one by one sequentially. If tested simultaneously, two highly correlated characteristics could yield to assigning weights almost randomly to either of them, thus blurring the results. My concern goes especially to the relationship between in-between centrality and eigenvector centrality. Is it why the authors are using ridge-regression?

2. The introduction states that "no single tract in isolation was necessary or sufficient to predict social network characteristics", however the authors haven't tested this assumption on the major white matter tracts that they have identified with Tracula, so we don't know if those tracts in isolation could predict social network characteristics. It would be interesting to have the result of such analysis.

3. One limitation of the design that cannot be overcome by the manuscript, but should be acknowledged in the Results and/or Methods sections, and could be discussed in the Discussion, is the use of ROIs which are not functionally identified for each subject, but are derived from group-level analysis instead. This approximation could be problematic for ROI-based analysis because of the idiosyncratic location in the brain of e.g. TPJ, or face areas, which are abutted by areas with opposite functional computation, object areas or body areas, etc (Fedorenko, Hsieh et al. 2010). While it is possible that this limitation might not change positive results, it could possibly have an impact on negative results by adding noise to the analyses from non-socially specific tracts.

4. Does the size of ROIs inherently bias FA? It is a candid question, as I don't know the answer to it. And if so, how is it taken into account at the group level or subject level: Would socio-affective networks with larger ROIs result in better FA prediction for this socio-affective network? Would subjects more central have larger ROIs that would make better FA predictions?

Minor points:

Abstract:

5. I find it pretty remarkable to be able to predict ones' social position with white matter tracts. Although correlational, the result is not trivial. The direction of causality between white matter connectivity and social position, however, is not at the core of the paper. To my opinion, a statement on heritability (genetic, structural, family-taught?) in the abstract ("driven by individual differences in the neuroanatomy") is not adding weight to the conclusion of the manuscript. It is more interesting to discuss this matter in the Discussion, as the authors are doing in the Future directions part rather than using it as a starting point for the manuscript. The wording is more neutral in the Introduction ("which may be reflected in individual differences in the networks of brain regions") than in the Abstract.

Results:

6. L96: "Characterize each cohort's social network". Could you please provide information about the 3 social networks: What is each network mean (and median) number of ties? What is each network density?

7. L110-113: Please provide a description of the ROIs obtained with Neurosynth for the 4 socio-affective networks, in order for the readers to compare present results with existing literature. Information about locations of each ROI and brain area it lies in would be welcome in main or supplementary text. Please also mention here that this Method provides an approximation of where the ROI would have been if it had been mapped with fMRI in each individual (see main concern).

8. L123-125: Please provide a description of the tracts, in order for the readers to compare present results with existing literature. Information about the 33+50+41+49 white matter tracts identified, e.g. which fasciculus they are part of, would be welcome as supplementary table.

9. What was the rationale in this study for not investigating tracts that link networks to each other (e.g. past work from the group(Parkinson and Wheatley 2012)) and focusing on within networks' tracts only?

10. L204: "We observed similar results". The variables predicted are not the same. A more precise wording would be more accurate.

11. L250: "The microstructural integrity of single tracts was not predictive of any of the five social network position characteristics". This would need to be tested also on the main tracts to reach the conclusion in the abstract.

Methods:

12. L502: "3-4 months after the subjects arrived on campus". What was the delay between social network assessment and scanning? Are the networks stable in this period of time?

13. L542: I think "of" is missing in "by the eigenvector of A".

14. L596: "in the event of asymmetric relations, only incoming rather than outgoing ties were used to compute eigenvector centrality". Are symmetric relations counted double?

15. L548-550. Betweenness centrality: Is it calculated on a directed or undirected graph?

16. L557. Constraint. "the presence of any social tie, irrespective of its direction was used to compute the constraint.". Does incoming tie + outgoing tie = 1 or incoming tie + outgoing tie = 2?

17. L574 "with a P" word missing. How many volumes per subject?

18. L603: Please mention here that it provides an approximation of where the ROI would have

been if it had been mapped in each individual (see main concern).

19. L605: Is the dilation by one voxel performed to reach white matter from the grey matter?

20. L613-615: "For each proposed[...]analysis". Not sure I understand what is done here

21. L630: Please indicate across what the FA values are averaged.

22. L642: What is the StandardScaler function doing?

23. L679: Is it 3 or 2 models?

24. L688: Which data analytic procedure is it referring to?

25. L717: Is a fold referring to a tract or a fasciculus?

Figures

26. Figure1: Please indicate which method is used for nodes positioning in these graphs.

27. Figure 2: Panel c is not showing the actual mirroring network, unless you place the red brain regions as in Panel b

28. Figure 3: are the social characteristics tested together? If so the figure is not capturing it.

29. Figure4: Is each point a subject? Why are the variables varying between [-4 3] and not [-1 1] (In the text it is indicated that it is normalized)?

Supplementary information

30. L68-72: Is it why the authors are using ridge regression?

Fedorenko, E., P.-J. Hsieh, A. Nieto-Castañón, S. Whitfield-Gabrieli and N. Kanwisher (2010). "New method for fMRI investigations of language: defining ROIs functionally in individual subjects." *Journal of neurophysiology* 104(2): 1177-1194.

Parkinson, C. and T. Wheatley (2012). "Relating Anatomical and Social Connectivity: White Matter Microstructure Predicts Emotional Empathy." *Cerebral Cortex* 24(3): 614-625.

Wang, Y. and I. R. Olson (2018). "The Original Social Network: White Matter and Social Cognition." *Trends in Cognitive Sciences* 22(6): 504-516.

Comments from Editor

E1.1 Please address all points raised by reviewers. Please pay close attention to the comments regarding statistics by reviewer 1. Indeed, while the association is significant the claim of prediction would require predicting data out of sample.

Thank you for securing such thoughtful reviews and for the opportunity to address the concerns raised by reviewers. As detailed in this point-by-point response, we have revised the manuscript to address the feedback that we received, including the comments regarding statistics from Reviewer 1 (see response R1.2 below). We believe that the manuscript is now stronger as a result of incorporating this feedback.

Comments from Reviewer 1

R1.1 In this paper, the authors employed diffusion MRI to study brain network features (i.e., white matter connectivity) that contribute to social network characteristics (i.e., individual differences in social network positions). They found distributed patterns of white matter microstructural integrity, especially those supporting social brain networks and major white matter tracts, are predictive of real-world social network position characteristics. This is an elegantly simple study that has important implications for the 'brain network underlying social network' literature. The study is well-designed and has collected enormous multimodal data (behavioral + DTI); the results are clear, consistent with previous studies; and the conclusions they draw are well-justified from their results. While I enjoy reading this paper very much and recommend publication, but would like to see some changes first (mainly clarifications and discussions).

Thank you for your thoughtful comments and encouraging feedback. As detailed in this document, we have worked to address all of the comments that you raised, and think that doing so has significantly improved the manuscript.

R1.2 The authors collected data from three different cohorts. While the distributed white matter microstructure models are very impressive to predict real-world stuff, I am concerned with the generalizability of the models. Is it possible to train a model based on one cohort and then test on another? The theoretical question here is that: do all three cohorts share the same neural mechanism? I know this remark might be too harsh and the sample size in each cohort might not be enough for decent machine learning modelling, but at least the authors should discuss them for future directions.

Thank you for raising this issue. As alluded to near the end of your comment, the decision not to leave entire cohorts out at a time in our data-folding procedure stemmed from the fact that training a model based on one cohort and testing it on another cohort would result in too little training data. Indeed, this motivated our use of leave-one-out cross-validation to maximize the amount of training data that is fed into the model so as to minimize the amount of variance in model performance across data folds. We have included in the Conclusions and Future Directions section further commentary on how future research with larger samples would benefit

from training and testing models on completely separate samples of participants on p.15, lines 500-504:

“Furthermore, given a large enough sample size, future work would also benefit from training and testing models on different samples of individuals, as this would shed light on whether relationships between white matter microstructural integrity and social network position characteristics are consistent across contexts and communities.”

We note that, for each participant, the model used to generate their predicted social network characteristics was trained on data from a separate set of participants (i.e., training and testing data were always independent of one another). While the current results can thus be considered an example of out-of-sample predictive modeling, a complementary but distinct empirical question could be addressed using out-of-*cohort* predictive modeling -- i.e., to what extent are the same neural features associated with particular social network characteristics across different contexts/communities? While it is not feasible, given the sample size limitations noted above, to repeat our analysis by training and testing the model on separate cohorts of participants, we have included below an additional set of figures to begin to explore this question. For each of the examined brain networks (i.e., the affective processing, mirroring, mentalizing, and face perception networks), we calculated the correlation between each predictor (i.e., mean FA extracted from a given WM tract) and each social network characteristic. This was performed separately within each cohort. The resulting correlation heatmaps are visualized below (Figs. R1-R4); cases where FA values in a particular brain network were predictive of a given social network position characteristic are demarcated with a black rectangle.

By visually comparing values in different columns (i.e., across cohorts) for each row (i.e., for each tract) in these heatmaps, it is possible to evaluate the relative consistency in the relationships between predictors and a given social network position characteristic across the three cohorts. For example, there appears to be relatively more consistency across cohorts in the relative magnitudes of associations between social network position characteristics and FA values for WM tracts in the affective processing network (Fig. R1), whereas associations were relatively more heterogeneous across cohorts for tracts in the mentalizing network (Fig. R3). However, given that the sample size within a given cohort is relatively small ($N_{\text{cohort-1}} = 46$; $N_{\text{cohort-2}} = 32$; $N_{\text{cohort-3}} = 34$), these correlation values should be interpreted with caution.

Figure R1. Relationships between white matter microstructure in tracts in the affective processing network and social network position characteristics.

Figure R2. Relationships between white matter microstructure in tracts in the mirroring network and social network position characteristics.

Figure R3. Relationships between white matter microstructure in tracts in the mentalizing network and social network position characteristics.

Figure R4. Relationships between white matter microstructure in tracts in the face processing network and social network position characteristics.

R1.3 Page 4, line 96. A subset of subjects underwent DTI scans. Did author strategically choose those DTI subjects based on their critical network positions (optimal selection) or just randomly collected data from any subjects willing to be scanned. Please clarify.

Participants were not strategically selected based on their social network characteristics. We have added this additional detail in the main text in the Methods section on p.19, lines 644-646:

“The neuroimaging study was advertised to all students in each cohort via email. All students who were interested in participating and who passed a standard MRI safety screening participated in the DTI scan.”

R1.4 Page 4, line 107. The definition and meaning of the network metric 'constraint' is unclear. Does 'constraint' mean the extent to which one connects to unconnected/isolated individuals? The description is a bit of cursory and obscure.

We apologize that this was not sufficiently clear and we thank Reviewer 1 for bringing this to our attention. We have now added additional explanation of constraint earlier on in the main text (i.e., in the portion of the Results section referenced here) on p. 4 on lines 113-128:

“Thus, two measures of brokerage were considered: constraint and betweenness centrality. We examined betweenness centrality to be consistent with prior work that used betweenness centrality as a measure of brokerage in establishing the heritability of this aspect of social network position¹². We included constraint following our own prior work integrating neuroimaging and social network data²², and given that constraint is a more local measure of brokerage that may be more impacted by individuals’ own sociobehavioral tendencies, rather than those of other nearby individuals in the network²³. In contrast, betweenness centrality captures how often an individual lies on the shortest path between other people in the social network, and thus can be dramatically impacted by factors beyond an individual’s own sociobehavioral tendencies²³. For example, individuals with high betweenness centrality may not function as true brokers, as they may lie on the shortest path between others because they are merely in close proximity to a true broker²³. Thus, relative to betweenness centrality, constraint may bear a stronger relationship to the sociobehavioral tendencies that are characteristic of brokers, which in turn are reflected in patterns of white matter microstructural integrity in the brain.”

R1.5 In addition, what's the statistical relationship between five social network position characteristics (i.e., out-degree centrality, in-degree centrality, eigenvector centrality, betweenness centrality, and constraint) in the current behavioral data? I know these metrics are conceptually different, but it will be helpful for readers to see how real data speaks. The supplementary material only provides t-test, but no correlations across all subjects.

We have now included in the Supplementary Information a table (Table S1) showing the correlations between out-degree centrality, in-degree centrality, eigenvector centrality, betweenness centrality, and constraint. As one would expect, these metrics are related to one another, given, for example, that someone with very few friends would have small values for all social network centrality measures.

R1.6 Page 12, line 372. It is still possible that other types of white matter microstructure (e.g., number of streamlines, MD, AD, RD, or even FA in certain segments of a tract) that can predict out-degree centrality, in-degree centrality, and betweenness centrality. The authors might not need to run extra analyses but needs to mention this limitation in the discussion.

We have now included in the Conclusion and Future Directions section discussion of the benefits of future work testing if other types of WM-based features are predictive of social network characteristics on p.15 on lines 505-508:

“...future work may benefit from testing if other measures of white matter microstructure, such as axial diffusivity, mean diffusivity, radial diffusivity, or the number of probabilistic tractography streamlines are predictive of social network position characteristics.”

R1.7 Page 13, line 414. The way to define mentalizing network using Neurosynth in this paper can be inaccurate, as a recent study suggested that the spatial specificity of mentalizing network is highly heterogeneous across subjects (see Wang et al., 2021 NeuroImage; doi: 10.1016/j.neuroimage.2021.118115). This limitation could be discussed, as it might contribute to the null results on mentalizing network, and future research should be encouraged to use more optimal ways (i.e., functional localizer) to validate the null results.

Thank you for bringing this to our attention. We have now included discussion of the limitations of using meta-analysis-derived ROIs in the Discussion section on p.15 on lines 470-478:

“Furthermore, a limitation of the current study is that a group-level meta-analytic map was used to define regions of interest associated with the mentalizing network. However, recent work has demonstrated that the spatial specificity of the mentalizing network is highly heterogeneous across individuals⁷⁸. This finding may explain the current study’s null results when using patterns of white matter microstructural integrity within the mentalizing network to predict social network position characteristics, and future work may benefit from using functional localizers to identify mentalizing regions of interest that would be used as seed regions in probabilistic tractography.”

R1.8 Page 18, line 596. The author could improve the paper by providing more information on the ROI definition. For each social brain network, how many ROIs were selected from Neurosynth maps? What were their anatomical labels? How many voxels in each ROI, and how many ROIs were bilateral or unilateral. Was tractography implemented only among the same hemisphere ROIs or included cross-hemisphere ROIs. A table listing above information will be extremely helpful for open science.

As suggested, we have included in the Supplementary Information an additional table (Tables S2-S5) detailing the ROIs in each brain system. For each ROI, the corresponding voxel count was calculated by averaging the ROI’s voxel size across all participants in native diffusion space. We have also added more details to the “ROI definition” and sub-section of the Methods section to clarify this aspect of the data analytic procedure p. 20 on lines 681-690:

“This method provides an approximation of the location of a given ROI if it had been mapped with fMRI in each individual. Each ROI was then dilated by a single voxel and was masked using a brain mask in order to create more liberal ROI masks to ensure that ROIs would extend into neighboring white matter. Due to being exceptionally large and/or spanning multiple regions, a small subset of ROIs were then masked using subject-specific anatomical masks generated using the Freesurfer anatomical parcellation algorithm^{92–94} in order to split the ROI into smaller constituent ROIs, which were subsequently used for analysis. Details regarding the size of each ROI, the anatomical regions associated with each ROI, and whether the ROI was masked using subject-specific anatomical masks are provided in Tables S2-S5.”

Tractography was implemented only between ROIs in the same hemisphere, as is stated on p. 20 in the Methods section. We have now added additional detail to the “Probabilistic tractography” and sub-section of the Methods section to make this more clear p. 20 on lines 693-694:

“For each brain network, the FSL *probtrackx2* command was then used to perform probabilistic tractography between every possible pair of ROIs within each of the left and right hemispheres (i.e., tractography was not performed to trace inter-hemispheric tracts), with a contralateral hemisphere exclusion mask and a brainstem exclusion mask.”

Comments from Reviewer 2

R2.1 Investigating the contribution of white matter connectivity to social behavior is essential to understand the neural basis of our social self in health and disease. This manuscript takes on the extra challenge to investigate the link between white matter connectivity and one’s social position in real-world networks, a critical question to understand the link between white matter integrity and real-world behaviors. The study is building upon a beautiful and large dataset of neuroimaging scans from 3 independent cohorts of graduate students, which social networks have been fully sociologically characterized. Building upon previous work (e.g.(Wang and Olson 2018)), the authors studied white matter integrity by considering fractional anisotropy not only within major white matter tracts, but also within fasciculi identified for social cognition. By using machine-learning, they discover that neuroanatomy of white matter parallels one’s position within its’ social network in terms of eigenvector centrality, and brokerage, two sociological variables of the network. Please find below a detailed description of questions that could be addressed by the authors.

Thank you for your thoughtful comments and encouraging feedback. As detailed in this document, we have worked to address all of the comments that you raised, and think that doing so has significantly improved the manuscript.

R2.2 My main concern, which could be overcome by providing additional information and analyses, goes towards quantification of relationship in-between social network position characteristics; and how this is impacting on the model. Statistical tests would be needed to investigate correlation between each social network position characteristic (in-degree centrality, out-degree-centrality, eigenvector centrality, constraint, betweenness) with one other. It is unclear from the Methods section (L662) if the social characteristics are tested simultaneously by the model or if they are tested one by one sequentially. If tested simultaneously, two highly correlated characteristics could yield to assigning weights almost randomly to either of them, thus blurring the results. My concern goes especially to the relationship between in-between centrality and eigenvector centrality. Is it why the authors are using ridge-regression?

We have now included in the Supplementary Information a table (Table S1) showing the correlations between out-degree centrality, in-degree centrality, eigenvector centrality, betweenness centrality, and constraint. These metrics are closely related to one another, as one might expect, given, for example, that a person with very few friends would have small values for all social network centrality measures.

To clarify, social network characteristics were never used as predictors in any of the models. The FA values extracted from WM tracts were used as predictors to predict each social network characteristic, separately, rather than simultaneously. Thus, there is no issue of weights being randomly assigned to one of the multiple correlated social network position characteristics, since these were not the predictor variables. We have added clarification in the Methods section p. 21 on lines 722-725 to address this confusion:

“As described in the main text, we tested if inter-individual variability in patterns of white matter microstructural integrity would be predictive of individuals’ positions in their real-world social network (Fig. 3). **That is, the independent variables consisted of a multivariate set of predictors measuring white matter microstructural integrity, and the outcome variable consisted of a given social network position characteristic.**”

That said, it is true that the predictors in our models (i.e., the FA values for different tracts within each examined brain system) were unlikely to be fully independent of one another. Therefore, we used ridge regression due to multicollinearity among the predictors, which were mean FA values extracted from different WM tracts. We have added additional details in the Methods section to clarify this point p. 21 on lines 731-732:

“...Given the multicollinearity among the predictors, regularized ridge regression was used.”

R2.3 The introduction states that “no single tract in isolation was necessary or sufficient to predict social network characteristics”, however the authors haven’t tested this assumption on the major white matter tracts that they have identified with Tracula, so we don’t know if those tracts in isolation could predict social network characteristics. It would be interesting to have the result of such analysis.

As requested, we conducted additional analyses testing if mean FA values extracted from single WM tracts defined using TRACULA were predictive of any of the social network characteristics. These analyses returned null results. Additionally, we also tested if the exclusion of any one single predictor would significantly diminish the performance of the full model, which uses all 18 TRACULA-derived predictors. This analysis also returned null results.

On p. 9 (lines 266-267) and on p.10 (lines 281-282) of the revised manuscript, we have added the following statement in both the “No single white matter tracts are necessary to predict social network position characteristics” and “Single white matter tracts alone are not sufficient to

predict social network position characteristics” sub-sections of the Results section in order to convey the results of these analyses:

“This procedure was also repeated using the set of predictors derived from the TRACULA analysis, and this analysis also returned null results.”

R2.4 One limitation of the design that cannot be overcome by the manuscript, but should be acknowledged in the Results and/or Methods sections, and could be discussed in the Discussion, is the use of ROIs which are not functionally identified for each subject, but are derived from group-level analysis instead. This approximation could be problematic for ROI-based analysis because of the idiosyncratic location in the brain of e.g. TPJ, or face areas, which are abutted by areas with opposite functional computation, object areas or body areas, etc (Fedorenko, Hsieh et al. 2010). While it is possible that this limitation might not change positive results, it could possibly have an impact on negative results by adding noise to the analyses from non-socially specific tracts.

We thank the reviewer for pointing out these valid limitations of the methodology used in the current study. We have now included discussion of these limitations in the Discussion section p. 15 on lines 470-478:

“Furthermore, a limitation of the current study is that a group-level meta-analytic map was used to define regions of interest associated with the mentalizing network. However, recent work has demonstrated that the spatial specificity of the mentalizing network is highly heterogeneous across individuals⁷⁸. This finding may explain the current study’s null results when using patterns of white matter microstructural integrity within the mentalizing network to predict social network position characteristics, and future work may benefit from using functional localizers to identify mentalizing regions of interest that would be used as seed regions in probabilistic tractography.”

R2.5 Does the size of ROIs inherently bias FA? It is a candid question, as I don’t know the answer to it. And if so, how is it taken into account at the group level or subject level: Would socio-affective networks with larger ROIs result in better FA prediction for this socio-affective network? Would subjects more central have larger ROIs that would make better FA predictions?

In response to separate requests from both reviewers (see R1.8 earlier in this document and R2.8 later in this document), we have included in the Supplementary Information a set of additional tables detailing the ROIs in each brain system, including the sizes of each seed region (Tables S2-S5). There did not appear to be a systematic relationship between average ROI size within a brain network and the extent to which patterns of FA values (distributed across tracts within that network) were predictive of social network position characteristics. For example, of the brain networks examined in the current study, the affective processing network had the smallest mean ROI size and the mirroring network had the largest mean ROI size. At the same time, patterns of FA values in these two brain networks were most consistently

predictive of social network position characteristics. However, we note that the current study was not designed to answer this question and that future studies may shed more light on the relationship between the size of seed ROIs and the extent to which FA values within tracts between those ROIs are predictive of behavioral and cognitive traits.

Regarding the subject-level question noted at the end of this point: In the current study, we do not have the data to directly answer if more central subjects would have larger ROIs that would make better FA predictions, given that the ROIs in the current study were functionally defined based on other subjects' data (i.e., based on automated large-scale meta-analyses implemented in Neurosynth). However, future work using functional localizers to identify ROIs can examine if subject-level differences in ROI sizes can impact FA-based predictions of social network position characteristics. We have now included discussion of this point on p.16 of the Conclusions and Future Directions section on lines 509-517:

“Future work may also benefit from using functional localizers to identify seed regions of interest, as the location of functionally defined brain regions may vary from person to person⁷⁹. In particular, aspects of the mentalizing network have recently been shown to vary substantially across individuals⁷⁸. Thus, using functional localizers to define subject-specific seed regions of interest would likely confer greater sensitivity for detecting systematic links between patterns of white matter microstructural integrity and traits such as social network position characteristics. Furthermore, with functionally localized subject-specific ROIs, it is also possible to test if the size of ROIs may systematically affect FA-based prediction of social network position characteristics.”

Minor points - Abstract:

R2.6 I find it pretty remarkable to be able to predict ones' social position with white matter tracts. Although correlational, the result is not trivial. The direction of causality between white matter connectivity and social position, however, is not at the core of the paper. To my opinion, a statement on heritability (genetic, structural, family-taught?) in the abstract (“driven by individual differences in the neuroanatomy”) is not adding weight to the conclusion of the manuscript. It is more interesting to discuss this matter in the Discussion, as the authors are doing in the Future directions part rather than using it as a starting point for the manuscript. The wording is more neutral in the Introduction (“which may be reflected in individual differences in the networks of brain regions”) than in the Abstract.

We thank the reviewer for bringing this to our attention. Our initial motivation for including this wording in the abstract was to help convey that social network position characteristics are thought to be trait-level individual difference variables. However, Reviewer 2 brings up an excellent point, and we have revised the abstract accordingly by removing the reference to heritability in the second sentence. The second sentence of the abstract now reads on line 15:

“Human behavior is embedded in social networks. Certain characteristics of the positions that people occupy within these networks appear to **be stable** within individuals.”

Minor points - Results:

R2.7 L96: “Characterize each cohort’s social network”. Could you please provide information about the 3 social networks: What is each network mean (and median) number of ties? What is each network density?

We have added this information to the Methods section on p.16 on lines 590-596:

“For descriptive purposes, for each cohort’s social network, we then calculated the mean and median numbers of social ties across individuals (i.e., average total degree centrality, summing incoming and outgoing ties for each individual) and the reciprocity of the graph, which refers to the probability that person i nominated person j as a friend if person j nominated person i as a friend (mean social ties_{Cohort-1} = 91, median social ties_{Cohort-1} = 77, reciprocity_{Cohort-1} = 0.53; mean social ties_{Cohort-1} = 78, median social ties_{Cohort-1} = 70, reciprocity_{Cohort-1} = 0.49; mean social ties_{Cohort-1} = 55, median social ties_{Cohort-1} = 46, reciprocity_{Cohort-1} = 0.48).”

R2.8 L110-113: Please provide a description of the ROIs obtained with Neurosynth for the 4 socio-affective networks, in order for the readers to compare present results with existing literature. Information about locations of each ROI and brain area it lies in would be welcome in main or supplementary text. Please also mention here that this Method provides an approximation of where the ROI would have been if it had been mapped with fMRI in each individual (see main concern).

In the Supplementary Information, we have now included tables (Tables S2-S5) containing the requested information about each ROI. In the Results section, we now mention the limitation of the current method of defining ROIs on p. 4 on lines 132-133:

“Regions of interest (ROIs) in four *a priori* defined brain networks associated with different aspects of socio-affective processing (i.e., affective processing, face perception, mentalizing, and mirroring networks) were functionally defined using the meta-analysis tool Neurosynth²⁴ (Fig. 2). This method provides an approximation of the location of a given ROI if it had been mapped with fMRI in each individual.”

We reiterate this point in the Discussion section on p. 16 on lines 509-517:

“Future work may also benefit from using functional localizers to identify seed regions of interest, as the location of functionally defined brain regions may vary from person to person⁷⁹. In particular, aspects of the mentalizing network have recently been shown to vary substantially across individuals⁷⁸. Thus, using functional localizers to define subject-specific seed regions of interest would likely confer greater sensitivity for detecting systematic links between patterns of white matter microstructural integrity and traits such as social network position characteristics.”

R2.9 L123-125: Please provide a description of the tracts, in order for the readers to compare present results with existing literature. Information about the 33+50+41+49 white matter tracts identified, e.g. which fasciculus they are part of, would be welcome as supplementary table.

In the Supplementary Information, we have now included a table containing details about the ROIs (Tables S2-S5), between which tracts were traced using probabilistic tractography. In the current study, it is not straightforward to map the white matter tracts to major identified tracts, as these are tracts between functionally defined areas that in some cases may encompass only a portion of a major tract or may not correspond to a single major tract.

R2.10 What was the rationale in this study for not investigating tracts that link networks to each other (e.g. past work from the group(Parkinson and Wheatley 2012)) and focusing on within networks' tracts only?

Whereas the current work uses probabilistic tractography, the past work referenced in this comment used tract-based spatial statistics (TBSS), a different and more exploratory approach. While TBSS has the advantage of being largely automated, it has significant limitations -- for example, it can be difficult to confidently ascribe points in the white matter "skeleton" with particular white matter tracts. Thus, it is not straightforward to frame TBSS as a between-network analysis. We have now added additional text in the Discussion section that expands on our rationale for tracing white matter tracts only within brain networks on p.16 on lines 518-533:

"Additionally, the current work focused on patterns of white matter microstructure across tracts *within* brain networks, as it was motivated by an interest in linking individuals' social network position characteristics to anatomical connectivity among brain regions involved in particular mental functions (e.g., linking characteristics of tracts connecting brain regions involved in affective processing to measures of social network centrality). We did not test whether patterns of white matter microstructure across functionally defined tracts *between* brain networks were predictive of social network position characteristics. It is conceivable that patterns of white matter microstructure across between-network tracts and, correspondingly, interactions between different mental functions (to the extent that tracts between such networks support interactions between the corresponding mental functions), also play an important role in shaping social behavioral tendencies and therefore an individual's social network position characteristics. We did, however, find that patterns of white matter microstructure across TRACULA-defined major white matter tracts were predictive of social network position characteristics. Future work could benefit from specifically considering tracts between different functionally defined brain networks."

R2.11 L204: "We observed similar results". The variables predicted are not the same. A more precise wording would be more accurate.

We have adjusted the wording in the main text on page pp. 8-9 on lines 228-232:

“We observed results similar to those in the primary analysis (see Supplementary Information), such that patterns of microstructural integrity were predictive of eigenvector centrality and betweenness centrality, and were also predictive of in-degree centrality and eigenvector centrality when controlling for demographic characteristics (age, gender), extraversion, handedness, and cohort.”

R2.12 L250: “The microstructural integrity of single tracts was not predictive of any of the five social network position characteristics”. This would need to be tested also on the main tracts to reach the conclusion in the abstract.

As requested, we conducted additional analyses testing if mean FA values extracted from single WM tracts defined using TRACULA were predictive of any of the social network characteristics. These analyses returned null results. Additionally, we also tested if the exclusion of any one single predictor would significantly diminish the performance of the full model, which uses all 18 TRACULA-derived predictors. This analysis also returned null results. As noted in our response to point R2.3 earlier in this document, we have added these details to the main text in the Results section on p. 9 (lines 266-267) and p.10 (lines 281-282):

“This procedure was also repeated using the set of predictors derived from the TRACULA analysis, and this analysis also returned null results.”

Minor points - Methods:

R2.13 L502: “3-4 months after the subjects arrived on campus”. What was the delay between social network assessment and scanning? Are the networks stable in this period of time?

We have now added these additional details in the main text in the Methods section on p. 19 on lines 637-640):

“A subset of 130 individuals who had completed Part 1 of the study completed a subsequent dMRI study ($n_{Cohort-1} = 54$; $n_{Cohort-2} = 36$; $n_{Cohort-3} = 40$). Subjects in cohort 1 were scanned 6-7 months after their arrival on campus, subjects in cohort 2 were scanned 7-8 months after their arrival on campus, and subjects in cohort 3 were scanned within the first month of their arrival on campus.”

We do not have multiple waves of data for all cohorts in this sample, and thus we are unable to assess if the social networks remain stable across time. However, the nature of this graduate program is very immersive and allows the participants to form intense and stable bonds relatively quickly, and they are assigned to groups that structure their everyday interaction partners with whom they take classes and collaborate on projects early on in the program.

R2.13 L542: I think “of” is missing in “by the eigenvector of A”.

This correction has been added to the main text in the Methods section on page p. 18 on line 609:

“The eigenvector centrality of each node is given by the eigenvector of **A** in which all elements are positive.”

R2.14 L596: “in the event of asymmetric relations, only incoming rather than outgoing ties were used to compute eigenvector centrality”. Are symmetric relations counted double?

In line with common practices in social network analysis, only incoming ties were used for calculating eigenvector centrality. We would be happy to expand on this point in the main text if it would be clarifying.

R2.15 L548-550. Betweenness centrality: Is it calculated on a directed or undirected graph?

Betweenness centrality was calculated using undirected graphs. We have added this detail to the main text in the Methods section on page p. 18 on lines 617-618:

“The betweenness centrality of an individual was calculated as the proportion of shortest paths between two given nodes that pass through the individual. **An unweighted, undirected graph was used to estimate betweenness centrality.**”

R2.16 L557. Constraint. “the presence of any social tie, irrespective of its direction was used to compute the constraint.”. Does incoming tie + outgoing tie = 1 or incoming tie + outgoing tie = 2?

Because constraint was calculated using unweighted, undirected graphs, here, an incoming tie + outgoing tie = 1. In other words, an edge between two nodes, A and B, would be treated equivalently in the computation of constraint regardless of whether both A and B nominated one another as friends, or if only one of these individuals reported the other as a friend. We have rephrased the relevant sentence to clarify this point on p. 19 on line 626:

“An unweighted, undirected graph was used to estimate constraint; that is, the presence of any social tie, irrespective of its direction **or if it was reciprocated**, was used to compute the constraint of each node.”

R2.17 L574 “with a P” word missing. How many volumes per subject?

This typo has been corrected, and we have added additional text explaining that we collected 33 volumes per subject in the Methods section on p. 19 on lines 648-653:

“Magnetic resonance imaging was conducted with a **Philips Achieva 3.0 Tesla scanner using a 32-channel phased array head coil**. Diffusion-weighted images were collected

using 70 contiguous 2 mm thick axial slices with 32 diffusion directions (91 ms TE, 8845 TR, 1000 s/mm² b-value, 240 mm FOV, 90° flip angle, 1.875 mm x 1.875 mm x 2 mm voxel size). Thirty-three diffusion-weighted volumes were collected per subject.

R2.18 L603: Please mention here that it provides an approximation of where the ROI would have been if it had been mapped in each individual (see main concern).

As noted in response to points R2.4 and R2.8, this detail has been added to the Results section on page p. 4 and Methods section on page p. 20.

R2.19 L605: Is the dilation by one voxel performed to reach white matter from the grey matter?

This is correct; ROIs were dilated by one voxel to create more liberal ROIs that had a higher chance of containing white matter. This additional detail has been added to the Methods section on p. 20 on lines 683-684:

“Each ROI was then dilated by a single voxel and was masked using a brain mask in order to create more liberal ROI masks to ensure that ROIs would extend into neighboring white matter.”

R2.20 L613-615: “For each proposed[...]analysis”. Not sure I understand what is done here

We have added additional clarification on p. 20 on lines 699-700:

“For each proposed white matter tract, if more than half of the subjects yielded zero fiber tracts (i.e., if more than half of the subjects did not have a valid tracing), the corresponding white matter tract was excluded from further analysis.”

R2.21 L630: Please indicate across what the FA values are averaged.

FA values were averaged across voxels within each tract. We have clarified this in the “Probabilistic tractography” subsection of the Methods section on page p. 21 on lines 713-717:

“For each subject, FA values were extracted from each voxel from each white matter tract image using subject-specific FA maps thresholded at 0.2, which is a standard practice to create conservative FA maps. The resulting FA values were averaged across voxels within each tract to yield a single FA value reflecting the microstructural integrity of a given white matter tract.”

R2.22 L642: What is the StandardScaler function doing?

The StandardScaler function subtracts the mean and scales to unit variance. We have now added this detail to the Methods section on p. 21 on lines 726-731:

“Using Scikit-learn’s Pipeline function, we created an algorithm that performed two steps in sequence on the training data for each fold (models fit to each fold’s training data were used to predict social network position characteristics based on white matter microstructural integrity values in the corresponding testing data): (1) normalize the predictors using Scikit-learn’s StandardScaler function (which subtracts the mean and scales to unit variance) and (2) implement ridge regression.”

R2.23 L679: Is it 3 or 2 models?

There are 3 models being referenced here. One predicts constraint from patterns of WM microstructural integrity in the affective processing network, one predicts constraint from patterns of WM microstructural integrity in the mirroring network, and another predicts eigenvector centrality from patterns of WM microstructural integrity in the mirroring network. We have reworded the relevant sentence to clarify this point on page p. 22 of the revised manuscript on lines 768-770:

“The primary results demonstrate that patterns of microstructural integrity distributed across white matter tracts in the affective processing network are predictive of constraint, and that patterns of microstructural integrity distributed across white matter tracts in the mirroring network are predictive of constraint and eigenvector centrality, even while controlling for variables including age, gender, handedness, cohort, and extraversion. Thus, in these three models (two models predicting constraint; one model predicting eigenvector centrality), we tested if any predictors...”

R2.24 L688: Which data analytic procedure is it referring to?

This is referring to the permutation testing procedure used to generate null distributions against which model performance can be measured. We have now added this detail to the Methods section on p. 22 on lines 779-781:

“The permutation testing procedure used in the primary analysis was then used to calculate the performance of each of the P leave-one-tract-out models in predicting the relevant social network position characteristic.”

R2.25 L717: Is a fold referring to a tract or a fasciculus?

Here, a fold refers to a data fold used in cross-validation. To clarify this, we have replaced the term “fold” with the phrase “data fold” and now explicitly point the reader to the description of data folds in a preceding section. The revised text can be found on p. 23 of the main text on lines 807-812:

“For each statistical test, we used Scikit-learn’s Pipeline function to create an algorithm that performed two steps in sequence on the training data for each fold (models fit to each data fold’s training data were used to predict social network position characteristics

based on the white matter microstructural integrity value in the corresponding testing data; see “Structural connectome-based predictive modeling of social network position characteristics” for details of the data-folding procedure)...”

Minor points - Figures

R2.26 Figure 1: Please indicate which method is used for nodes positioning in these graphs.

We used the Fruchterman-Reingold layout algorithm as implemented in the igraph package (layout_with_fr). This detail has been added to the caption of Figure 1.

R2.27 Figure 2: Panel c is not showing the actual mirroring network, unless you place the red brain regions as in Panel b

Thank you for bringing up this issue. This figure is meant to be a schematic for reference and is not meant to actually visualize the mirroring network. We have changed the title of Panel c to reflect this and to avoid implying that this panel depicts the actual mirroring network, and we have also repositioned the red brain regions in Panel c to match the positions of the red clusters in Panel b. The revised version of Figure 2 is provided below for convenience:

R2.28 Figure 3: are the social characteristics tested together? If so the figure is not capturing it.

As noted in greater detail in response to point R2.2, social network position characteristics are the predicted variables and are not the predictor variables. Thus, they are predicted separately. As noted in response to point R2.2, we have revised the text to clarify this.

R2.29 Figure4: Is each point a subject? Why are the variables varying between [-4 3] and not [-1 1] (In the text it is indicated that it is normalized)?

Each point is a subject's predicted and actual social network position value. The values are z-scored such that the standard deviation is equal to one. The values are not scaled to a range of -1 to 1. Subject's social network position values were z-scored prior to analysis. This additional detail has been added to the caption of Figure 4 on p. 8 on lines 215-218:

“For each model, the predictive performance was measured using the Pearson correlation between the actual and predicted values of the social network variable of interest; *p*-values are FDR-corrected; shaded regions indicate 95% CIs. Social network position values were z-scored prior to analysis.”

Minor points - Supplementary Information

R2.30 L68-72: Is it why the authors are using ridge regression?

As noted in response to point R2.2, the predictors in our models were average FA values in the various tracts comprising each functional brain network. Thus, ridge regression was used because of likely multicollinearity among FA values corresponding to different tracts within each brain network. We have clarified the model specification and motivation for our analysis approach in the Methods section on p. 21 on lines 722-725 and lines 731-732:

“As described in the main text, we tested if inter-individual variability in patterns of white matter microstructural integrity would be predictive of individuals' positions in their real-world social network (Fig. 3). That is, the independent variables consisted of a multivariate set of predictors measuring white matter microstructural integrity, and the outcome variable consisted of a given social network position characteristic... Given the multicollinearity among the predictors, regularized ridge regression was used.”

Review 2 references

Fedorenko, E., P.-J. Hsieh, A. Nieto-Castañón, S. Whitfield-Gabrieli and N. Kanwisher (2010). "New method for fMRI investigations of language: defining ROIs functionally in individual subjects." *Journal of neurophysiology* 104(2): 1177-1194.

Parkinson, C. and T. Wheatley (2012). "Relating Anatomical and Social Connectivity: White Matter Microstructure Predicts Emotional Empathy." *Cerebral Cortex* 24(3): 614-625.

Wang, Y. and I. R. Olson (2018). "The Original Social Network: White Matter and Social Cognition." *Trends in Cognitive Sciences* 22(6): 504-516.

Julia Sliwa

REVIEWERS' COMMENTS:

Reviewer #1 (Remarks to the Author):

The authors have successfully addressed all of my concerns and suggestions. I have no further comments.

Reviewer #2 (Remarks to the Author):

I looked at the revised manuscript and the authors responses: all my concerns have been addressed and clarified. The results are novel and of interest to others in the community and the wider field. The work is convincing and the paper will influence thinking in the field. Statistical analysis is appropriate and valid, the level of detail provided ensures the ability of a researcher to reproduce the work.